

# Climate transition over the past two centuries revealed by lake Ebinur in Xinjiang, northwest China

Xiaotong Wei[1], Hanchao Jiang[1*], Hongyan Xu[1], Yumei Li[1,2], Wei Shi[1], Qiaoqiao Guo[1], Siqi Zhang[1]

[1]State Key Laboratory of Earthquake Dynamics, Institute of Geology, China Earthquake Administration, Beijing 100029, China
[2]Development Research Center of China Earthquake Administration, Beijing 100036, China

Correspondence to: Hanchao Jiang (hcjiang@ies.ac.cn)

**Abstract**

Global climate has undergone dramatic changes over the past 200 years, accurately identifying the climate transition and its controlling factors will help to address the risks posed by global warming and predict future climate trends. To clarify climate change over the past 200 years, detailed analyses of chronology, grain size, color reflectance ($L^*$, $a^*$) and carbon content (TOC, TIC) were conducted on a 200-year high resolution (~ 2 a) sedimentary record from lake Ebinur in Xinjiang, northwest China. The results show that the median grain size (Md) of lake sediments ranges from 5.5 μm to 9.9 μm, with a mean value of 7.0 μm. Multi-parameter analysis of grain size suggests that the sediments in lake Ebinur are mainly transported by wind, and there are two kinds of different sources and transport processes: the fine-grained sediments (< 20 μm) are background dust that was transported by long distance high-altitude suspension, while the coarse-grained sediments (> 20 μm) are local and regional dusts that were transported from short distances at low altitudes. Comparative analysis of multi-proxies including grain size、color reflectance and carbon content reveals that 1920 AD is the time point of climate transition in the past 200 years. In the early period (1816-1920 AD), the high C values indicate strong transport dynamics; the high proportion of ultrafine component indicates strong pedogenesis, combined with high organic carbon content and high $a^*$ values, it is inferred that the water vapor content is relatively higher. Overall, this period corresponds to the cold and wet climate. In the later period (1920-2019 AD), the proxies show opposite changes, which may reveal a warm and dry climate. Based on a comprehensive analysis of multiple drivers (i.e., solar radiation, greenhouse gases and volcanic eruption), we propose that the increase of solar irradiance in 1920 AD played a dominant role in the Asian climate transition, and that the gradual rise in the concentrations of greenhouse gases ($CO_2$ and $CH_4$) may have a positive feedback effect on the climate transition.

**Keywords:** lake Ebinur; arid Central Asia; climate transition; solar radiation; greenhouse gases



## 1 Introduction

The global climate has undergone a clear transition over the past 200 years: from the cold Little Ice Age (LIA) to the 20th century warming (Jacoby et al., 1996; Jones and Mann, 2004; Zhou et al., 2011; Bokuchava and Semenov, 2021). However, the timing of climate transition is still ambiguous, which makes it difficult to clarify the contribution of the driving mechanisms of warming, such as solar radiation, volcanic activity and the concentrations of anthropogenic-related greenhouse gases (e.g., $CO_2$, $CH_4$) (Hansen et al., 1981; Mann et al., 1998; Schmidt et al., 2011; Huber and Knutti, 2012). Therefore, it is very crucial to accurately identify the timing point of climate transition over the past 200 years, which is essential for a deeper understanding of the driving mechanism of climate warming. This will provide a theoretical basis for better coping with the risks of global warming and even predicting future climate trends.

Temperature records compiled from the Northern Hemisphere (NH) show that natural variability (solar radiation, volcanic activity) and human activity (greenhouse gases, i.e., $CO_2$、$CH_4$) have driven the warming since the 20th century (Overpeck et al., 1997; Mann et al., 1998). Although this view of warming is widely agreed, the temporal turning point of warming remains controversial due to the differences in the materials and methods used to reconstruct the temperature series (Zhang, 1991; Overpeck et al., 1997; Yang et al., 2002; Weckström et al., 2006; Ge et al., 2013). Zhang (1991) proposed that the LIA in China ended in the 1890s, mainly by reconstructing winter temperature series from historical literatures. And Wang et al (2001) found that the average temperature in the 20th century was 0.4 °C higher than the average temperature of the past 1200 years by the weighted average of the temperature series in 10 regions of China. Furthermore, the warming over the past 100 years also shows the characteristics of periodic warming, accompanied by the secondary cold-warm fluctuations (Wang and Gong, 2000; Chylek et al., 2006; Chen et al., 2009). At the same time, there are some extreme droughts accompanied by warming, which also have the characteristics of temporal and spatial differences and periodic (Zheng et al., 2006; Yang et al., 2010; Gou et al., 2014). For example, tree-ring records and historical literatures indicate that extreme drought conditions in northern China occurred in 1928-1932 AD (Liang et al., 2006; Fang et al., 2012), while tree-ring δD record from Kenya demonstrates that extreme drought in East Africa in the early 1920s (Krishnamurthy and Epstein, 1985). The timing of climate transition over the past 200 year and the periodicity of the warming are still unclear. More detailed, high-resolution climate data are thus needed to reveal the temporal turning point and characteristics of warming in order to better deal with the current global warming.

Xinjiang, which covers 1/6 of China's land area, is located on the interior of the continent and is a representative region of arid Central Asia. The climate of the region is mainly influenced by the westerly circulation, which is characterized by low precipitation, high temperatures and fragile ecosystems, making it very sensitive to climate change (Chen et al., 2009; Huang et al., 2017; Yao et al., 2022). This region is far away from the eastern region and less affected by the East Asian summer monsoon, showing significant climatic differences from the eastern monsoon regions (Aizen et al., 2001; Huang et al., 2013). In addition, numerous studies and literatures indicate that





human activity has not been the dominant factor in environment evolution in the
western region until the 1950s (Chen et al., 2006b; Ma et al., 2014; Xue et al., 2019),
while in the eastern region, human activity had irreversible impacts on the natural
environment as early as 2000 years ago (Chen et al., 2020a, 2020b). The arid Xinjiang
region is therefore the perfect location to study the timing of climate transition over the
past 200 years in the natural state.

Lakes, especially closed lakes in arid and semi-arid regions, are very sensitive to
environmental changes (Chen et al., 2008; Liu et al., 2008; Wu et al., 2009). Many
proxies in lake sediments are often used to reconstruct past environmental evolution,
such as grain size (Qiang et al., 2007; Jiang and Ding, 2010; An et al., 2012), color
reflectance (Ji et al., 2005; Jiang et al., 2007, 2008), pollen (Wang et al, 2013; Chen and
Liu, 2022), total organic carbon and total inorganic carbon (Xiao et al., 2006, 2008).
Generally, reliable interpretation of these proxies requires adequate knowledge of
sediment sources and processes (Jiang et al., 2016). For arid Xinjiang region, frequent
aeolian sand activities will import more coarse particulate matter into the lake
(Abuduwaili et al., 2008; Ma et al., 2016), complicating the sources and transport
mechanisms of lake sediments. Thus, provenance studies on lake sediments in dry
Xinjiang are necessary before the interpretation of proxies.

In this study, we presented a new 200-year environmental record based on detailed
analysis of grain size, color reflectance ($L^*$, $a^*$), carbon content (TOC, TIC) and
chronology from lake Ebinur in Xinjiang, northwest China. Our aim is to clarify the
provenance and transport mechanisms of lake sediments, and to further explore the
timing of climate transition over the past 200 years and its possible driving mechanisms
in the inland arid region.

**2 Geographic and geologic settings**
Lake Ebinur (44°54'-45°08' N, 82°35'-83°10' E), located in the arid region of
northwestern China, is a closed salty lake (Fig. 1). It is surrounded by mountains on the
southern, western, and northern sides, and connected to the Junggar Basin in the east
(Ge et al., 2016). The lake has a drainage basin area of 50,321 km$^2$, including 24,317
km$^2$ of mountainous area, 25,672 km$^2$ of plain area and 542 km$^2$ of lake area
(Abuduwaili and Mu, 2006; Ma et al., 2014). Lake Ebinur is supplied by Bo, Jing and
Kuitun River (Fig. 1), which are mainly recharged by glacier melting and precipitation
in the high-altitude area of Tianshan Mountain (Hu, 2004; Wang et al., 2013). The lake
has a maximum water depth of 3.5 m with a mean depth of 1.2 m. Total dissolved solids
salinity in the lake varies from 85 g L$^{-1}$ to 124 g L$^{-1}$ (Wu et al., 2009).



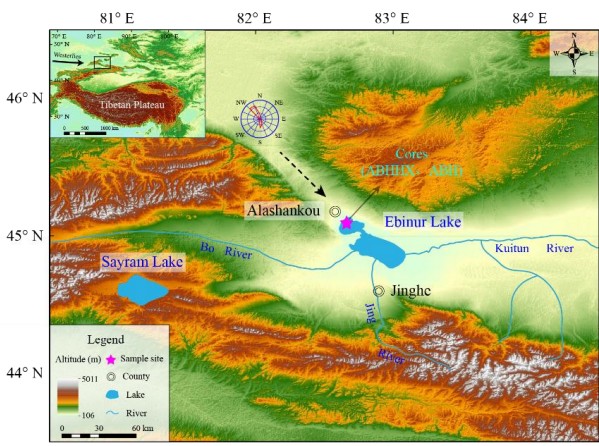


**Figure 1. Map showing the geomorphology, rivers, wind direction and sampling site of the study area.**


The climate of the study region is mainly dominated by westerlies and is a typical temperate continental climate, which is characterized by low rainfall and strong evaporation (Zhou et al., 2019, 2021). Data from the Alashankou meteorological station (45°11' N, 82°34' E) near Lake Ebinur show the mean annual temperature of 9.2 ℃, with a mean temperature of 27.9 ℃ in July and -14.9 ℃ in January (Fig. 2). The mean annual precipitation is 121 mm, and 63 % of the total precipitation occurs from May to September (Fig. 2). The mean annual relative humidity is about 53 %, with relative humidity exceeding 70 % in December, January, and February, and below 40 % in May-September (Fig. 2). The mean annual evaporation is 1315 mm, which is almost 10 times higher than the mean annual precipitation, resulting in an extremely arid climate (Zhou et al., 2019). The climate is generally characterized by warm-dry summers and cold-wet winters (Fig. 2). Temperate desert taxa dominate the modern vegetation types of the lake Ebinur region (Wang et al., 2013; Li et al., 2021), such as *Haloxylon*, *Tamarix*, *Ephedra*. Ala Mountain Pass, located in the northwest of lake Ebinur, is a well-known wind passage with a prevailing northwest wind all year round. The maximum wind speed can reach 55 m s$^{-1}$, with an average of 164 days per year when wind speed exceeds 20 m s$^{-1}$ (Wu et al., 2009; Ma et al., 2011). The unique topographic conditions of the region contribute to strong light, the frequent dust and salt dust storms (Abuduwaili et al., 2008). In addition, there are significant seasonal differences in potential dust transport pathways: longer transport distances in spring and summer, but shorter and lower transport distances in autumn and winter (Ge et al., 2016).

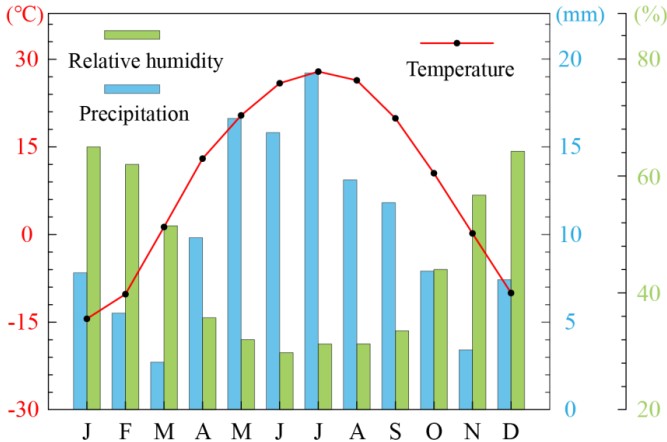

**Figure 2. Mean monthly temperature, mean monthly precipitation and mean monthly relative humidity in the lake Ebinur region. Data are the observational averages from 1981 to 2010 at Alashankou Meteorological Station (45°11′ N, 82°34′ E; 336.1 m a.s.l.; http://data.cma.cn/).**

## 3 Materials and Methods

In August 2019, two parallel sediment cores (ABHHX and ABH, of 48 cm and 50 cm length, respectively) were retrieved from the northwest edge of lake Ebinur at a water depth of 0.8 m (45°04' N, 82°36' E, 193 m), using a 60-mm UWITEC gravity corer (Fig. 1). Both cores were sampled consecutively in the field at 0.5 cm intervals, and 96 samples (ABHHX) and 100 samples (ABH) were obtained. Each sample was sealed in a separate plastic bag and taken back to the laboratory for analysis. Sediment samples from core ABHHX were used for chronology and multi-proxy analyses (grain size, color reflective, TOC and TIC), while samples from core ABH were used for chronology for comparison with ABHHX.

To construct the chronology of lake Ebinur sediments, the activities of $^{210}$Pb and $^{137}$Cs in the upper 20 cm of cores ABHHX and ABH were measured at 0.5 cm intervals by high purity Ge gamma spectrometer produced by EG company at the Nanjing Institute of Geography and Limnology, Chinese Academy of Sciences. Each dry sample was ground to < 100 mesh and sealed in a plastic tube for 3 weeks to achieve radioactive equilibrium (Appleby et al., 1986), and the measurement method followed Appleby (2001). The activity of total $^{210}$Pb ($^{210}$Pb$_{tot}$) was determined via gamma emissions at 44.5 keV, and the activity of $^{226}$Ra was determined by measuring the activity of its daughter nuclide $^{214}$Pb at 295 keV and 352 keV. The activity of $^{137}$Cs was measured with the 662 keV photopeak. The supported $^{210}$Pb activity was assumed to be in equilibrium with in situ $^{226}$Ra activity, and the unsupported $^{210}$Pb activity ($^{210}$Pb$_{ex}$) was calculated by subtracting the $^{226}$Ra activity from the $^{210}$Pb$_{tot}$ (Pratte et al., 2019).

A total of 96 samples were obtained from core ABHHX at 0.5 cm intervals, and the grain-size distribution of each sample was measured using a Malvern Mastersizer 3000 laser grain-size analyzer at the State Key Laboratory of Earthquake Dynamics, Institute of Geology, China Earthquake Administration. Approximately 0.2 g of the



dried sample was treated with 10 mL of 30 % $H_2O_2$ and 10 mL 10 % HCl to remove
the organic matter and carbonate. After the sample solutions were washed to neutral,
10 mL of 0.05 M $(NaPO_3)_6$ was added, and the mixed solutions were shaken for 10 min
in an ultrasonic vibrator to disperse the sample before analysis. The Mastersizer 3000
analyzer automatically outputs the volume percentage of each grain-size fraction, and
the measurement range is 0.01-3500 μm with a relative error of < 1 %.
96 samples from core ABHHX at 0.5 cm intervals were also used for the
measurements of color reflectance ($L^*$, $a^*$), TOC, and TIC. Each Sample of about 1.5 g
was dried at 40 ℃ for 24 h, then crushed without damaging their grain-size (Jiang et
al., 2008) and the color reflectance was measured by using a SPAD 503 handheld
spectrophotometer. For the measurement of carbon content, the samples were ground
into powder finer than 61 μm and dried at 40 °C for 24 h. The total carbon (TC) contents
of samples were first measured at 960 ℃ using an Elementar Rapid CS analyzer. Then
each sample was pretreated with 1 M HCl solution to remove carbonates, and TOC
content was measured (Fan et al., 2020). The dry samples were weighed before and
after carbonate removal, and the actual TOC values were obtained by converting the
measured TOC values using the ratio of the mass before and after treatment. The
difference between TC content and TOC content is TIC content. The relative error
analysis of carbon content is less than 1 %. These experiments were all conducted at
the State Key Laboratory of Earthquake Dynamics, Institute of Geology, China
Earthquake Administration.

**4 Results and Interpretation**
**4.1 Chronology**
Generally, the chronology of modern lake sediments is established by [210]Pb and
[137]Cs dating methods (Ma et al., 2015, 2016). The unsupported [210]Pb activity ([210]Pbex)
showed decreasing trend with depth until it stabilized at about 20 cm (Fig. 3c). The
[210]Pbex of core ABHHX varied from 6.6 Bq kg[-1] to 97.6 Bq kg[-1], while the [210]Pbex of
the core ABH decreased from 114.5 Bq kg[-1] at the surface to a minimum value of 6.8
Bq kg[-1] (Fig. 3c). According to the constant rate of supply (CRS) model (Appleby and
Oldfield, 1978), the core ABHHX was dated at 17.5 cm to 1944 AD, while the core
ABH was dated at 16 cm to 1940 AD. The calculated deposition rate by core ABHHX
is 2.33 mm yr[-1], which is very close to the rate of 2.03 mm yr[-1] from core ABH. However,
the [137]Cs activity of both cores is 0 (Fig. 3c), which may be related to the rapid decay
and downward migration of [137]Cs (Xiang, 1995; Gao et al., 2021). The core of lake
Sayram, ~ 120 km away from lake Ebinur, showed a deposition rate of 2.07 mm yr[-1]
calculated by CRS model since 1955 (Ma et al., 2015), further confirming the reliability
of the age. In addition, several climatic events revealed by this core are well consistent
with regional records (see Sect. 5.2). Accordingly, the 48 cm core was dated back to
1816 AD using a linear extrapolation method (Fig. 3c).





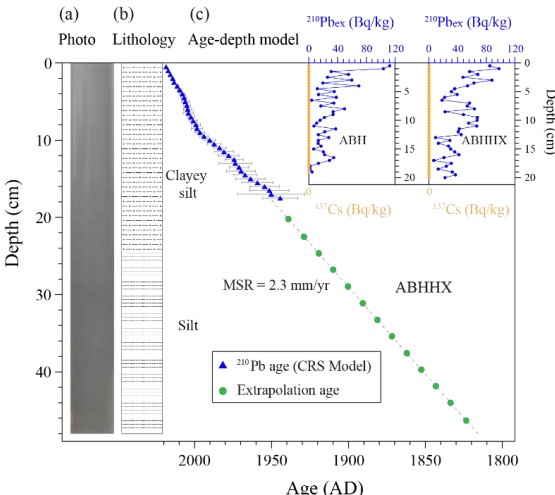

**Figure 3. (a) Photo and (b) lithology of the ABHHX core; (c) Age-Depth Model of the ABHHX**

**core. $^{210}Pb_{ex}$ activity and $^{137}Cs$ activity for the ABH core and the ABHHX core (upper right).**

## 4.2 Sedimentary Proxies record

The grain size composition of lake Ebinur sediments is dominated by fine grains (median grain size (Md): 5.4-9.9 μm, mean 7.0 μm) (Fig. 5a). All 96 grain-size data were analyzed by end-member analysis (EMA) using AnalySize software (Weltje, 1997; Paterson and Heslop, 2015; Jiang et al., 2022). The results show that the correlation coefficient ($r^2$) is as high as 0.98 when the number of end member is 2 (Fig. 4b). Overall, the lower part of the sedimentary sequence is characterized by large fluctuations and coarse grains (Figs. 5a-5g), with over 60 % of the C value above 50 μm (Figs. 6b, 6c), reflecting strong transport dynamics (Passega, 1964; Jiang et al., 2017a; Wei et al., 2021). The particles in the upper part of the sequence are finer (Figs. 5a-5g), and only ~ 20 % the C value exceeds 50 μm (Figs. 6d, 6e), indicating smaller transport dynamics. In addition, other proxies of lake Ebinur sequence ($L^*$, $a^*$, TOC and TIC) also show different variation characteristics in the corresponding upper and lower parts (Figs. 5k-5n). Thus, the sedimentary sequence can be divided into two units.

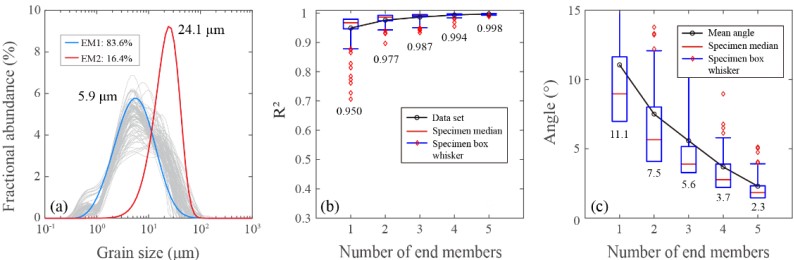

**Figure 4. End-member modeling results of the ABBHX core: (a) grain-size distribution for all**

**96 samples and two selected end–members; (b) correlation between the multiple correlation**

**coefficient ($R^2$) and the number of end members; (c) correlation between the angle and the**




**number of end members.**

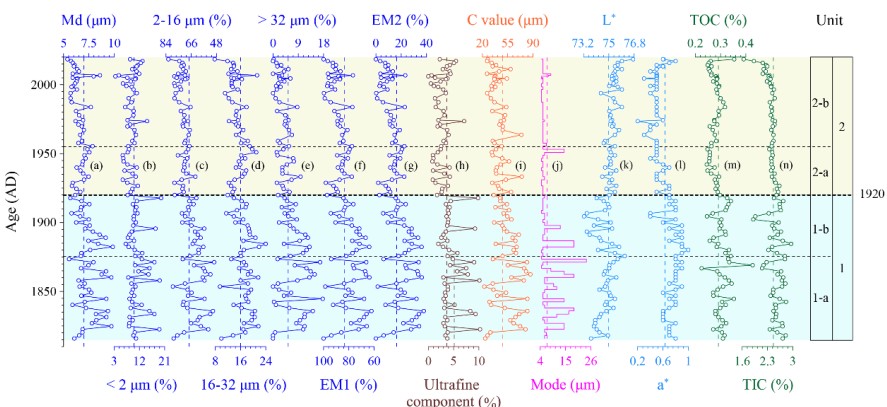


**Figure 5. Variations of grain size, color reflectance and carbon content for the ABHHX core:**
**(a) mean grain size; (b-g) percentages of < 2 μm, 2-16 μm, 16-32 μm, >32 μm fractions, EM1**
**and EM2; (h) the proportion of ultrafine component (< 1 μm fraction); (i) the one percent of**
**grain size (C value); (j) the modal size of grain size (Mode); (k) L$^*$; (l) a$^*$; (m) the total organic**
**carbon content (TOC); (n) the total inorganic carbon content (TIC).**

*Unit 1 (48-24 cm, 1816-1920 AD)*
During this period, all proxy records (grain size, reflectance, carbon content) are
characterized by large-amplitude fluctuations (Figs. 5a-5n). The grain size is the
coarsest in the whole sequence, with high-amplitude fluctuations (Md (median grain
size): 5.5-9.9 μm, mean 7.5 μm) (Fig. 5a). The variation of Md is clearly influenced by
the coarse component: EM2 (0-37.6 %, mean 21.6 %) (Fig. 5g), i.e., the coarse particles
are deposited first, and the fine particles are deposited later. Combined with the higher
C value for this unit (22.6-86.5 μm, mean 54.5 μm) (Fig. 5i), it indicates that the wind
is stronger at this stage, bringing more coarse-grained matter from local and regional
dust (see Sect. 5.1 for the explanation of provenance). This may mean that the
temperature is low and the wind transport is strong during this period, which is
consistent with the simulation result of Ge et al. (2016) that the winter transport in the
study area is mainly carried at low altitude and short distance. Correspondingly, the
contents of TOC (0.25-0.34, mean 0.30) and TIC (1.91-2.95, mean 2.50) also showed
strong fluctuations (Figs. 5m, 5n), which may have been influenced by strong wind
activity during the cold period.
In general, the ultrafine component (the grain size fraction of < 1 μm) is associated
with pedogenesis and can be used as indicator of regional climate change (Sun, 2006;
Sun et al., 2011). In this unit, the proportion of ultrafine component is the highest in the
whole sequence (1.7 %-10.2 %, mean 5.0 %), revealing the strongest pedogenesis in
the study area (Fig. 5h). It is generally believed that pedogenesis is related to
temperature and humidity (Sun et al., 2011). However, the temperature was lower and
the wind speed was higher during 1816-1920 AD, so we considered that the strong
pedogenesis during this period might be related to the high humidity. During the cold



LIA, the westerlies circulation brought more water vapor to the arid Central Asia (Chen
et al., 2010, 2015). In relatively humid climate, the pedogenesis of sediments is
enhanced, producing more fine-grained clay minerals (Deng et al., 2022; Sun, 2006).
a* is usually affected by red minerals (e.g., hematite and goethite) and is thought to be
associated with oxidation of sediments in arid region (Ji et al., 2005; Jiang et al., 2007).
The high a* value (mean 0.76) in this unit indicates that more water vapor enhanced the
oxidation during the cold period (Fig. 5l), thus providing more red minerals for the lake.
Related to humidity fluctuation, L* values within arid lakes are considered to reflect
variations in the carbonate, and high L* values denote more carbonate content (Xiao et
al., 2006; Jiang et al., 2008). The L* value in this unit fluctuates between 73.2-76.1,
with an average of 74.6 (Fig. 5k), which may be related to the changes of the lake water
body. The cooling leads to weakening of evaporation and transpiration, and together
with more water vapor from the westerlies (Guo et al., 2022), resulting in more water
in the lake and more carbonate content.

In summary, the climatic conditions in the study area were predominantly cold and
wet during 1816-1920 AD, which was consistent with cold and wet LIA in arid Central
Asia revealed by previous studies (Chen et al., 2006a; Chen et al., 2015). Further, this
unit can also be divided into two sub-units based on the changes of all proxies: unit 1-
a and unit 1-b (Figs. 5a-5n). During the period of unit 1-a (48-34 cm, 1816-1876 AD),
the study area has been under cold and wet climatic conditions, while unit 1-b (34-24
cm, 1876-1920 AD) was in a transition period from cold and warm.

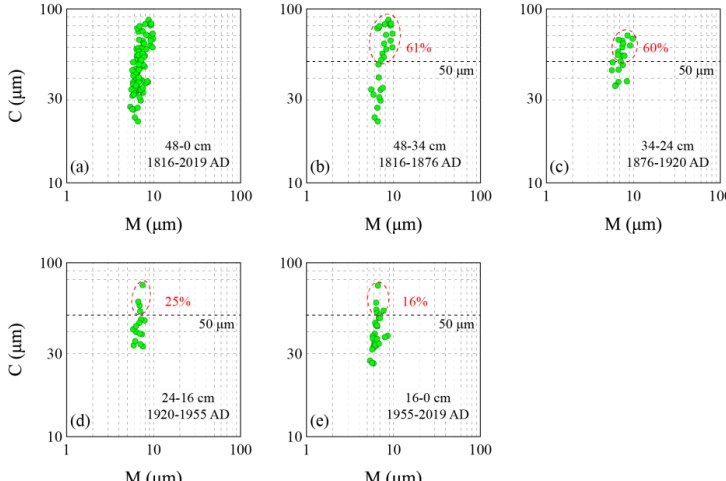

**Figure 6. C-M plot for: (a) all samples during 1816-2019 AD; (b-e) samples during four**
**different time intervals.**

***Unit 2 (24-0 cm, 1920-2019 AD)***
On the whole, the sedimentary proxies in this unit show a stable variation with
slight fluctuations (Figs. 5a-5n). The particle size is the finest in the whole sequence,
and Md varies from 5.4 μm to 8.6 μm with a mean value of 6.5 μm (Fig. 5a). Both of

Climate of the Past
2022 Author(s)
en



EM2 percentage (0-23.2 %, mean 11.3 %) and C value (26.4-76.5 µm, mean 41.9 µm)
decreased significantly (Figs. 5g, 5i), indicating a weakening of wind intensity and
lower coarse particles matter. This is probably duo to the decrease of the temperature
gradient as the temperature rise (Zhang et al., 2021), resulting in the weakening of wind
intensity and the decrease of coarse particles transported at low altitude and short
distance (Ge et al., 2016). As well, the contents of TOC (0.25-0.32) and TIC (2.17-2.63)
showed very slight fluctuations except for the top two points (Figs. 5m, 5n).
As shown in Figure 5, the proportion of ultrafine component during this period is
lower (0.8 %-4.3 %, mean 2.2 %), revealing weaker pedogenesis. The obvious decrease
of $a^*$ value (mean 0.51) indicates the weakening of oxidation (Fig. 5l), which may be
caused by reduced water vapor from westerly circulation and enhanced evaporation due
to the increase in temperature. And the relatively high $L^*$ value (mean 75.3) may be
associated with an increase in summer glacial meltwater into the lake as a result of
warming (Yao et al., 2022). However, the increase of $L^*$ value since 1955 AD may be
related to the dramatic shrinkage of lake Ebinur by human activity (see Sect. 5.2).
Thus, the changes of these proxies in this unit indicate that a warm and dry climate
in the study area during 1920-2019 AD. Similarly, unit 2 can be further divided into
two sub-units according to the variation of all proxies: unit 2-a (24-16 cm, 1920-1955
AD) and unit 2-b (16-0 cm, 1955-2019 AD) (Figs. 5a-5n).

**5. Discussion**
**5.1 Provenance and transport mechanisms of lake Ebinur sediments**
The Y value of Sahu's formula is usually used to recognize the eolian environment,
which is mainly determined by mean grain size, standard deviation, skewness, and
kurtosis (Sahu, 1964). The Y values of all samples range from -19.5 to -7.6, lower than
the threshold value of -2.74 (Fig. 7), supporting their windblown origin (Jiang et al.,
2017b, 2022; Wei et al., 2021). In addition, the arid and windy climate in the study area
also provides favorable conditions for aeolian deposition (Abuduwaili et al., 2008; Liu
et al., 2015; Ge et al., 2016). Previous studies have also shown that sediments in lakes
located in arid and windy areas may be transported by wind (Jiang et al.,2014; Wei et
al., 2021). These suggest that it is feasible for us to interpret the lake Ebinur sediments
as an aeolian source.

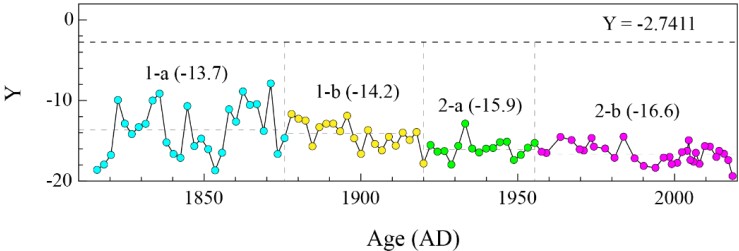

**Figure 7. The Y values for the ABHHX core, determined by the Sahu formula (Sahu, 1964).**

End-member simulations of all 96 grain size data show that there are two end-





member components in lake Ebinur sediments: EM1 (~ 5.9 μm) and EM2 (~ 24.1 μm)
(Fig. 4a). This is consistent with previous studies (Pye, 1987; Jiang et al., 2014; Wei et
al., 2021), i.e., the fine particles (EM1) are transported by long distance high-altitude
suspension and represent background deposition, while the coarse particles (EM2) are
transported by short distance low-altitude and represent local and regional deposition.
In addition, the EM2 component (~ 24.1 μm) shows a similar modal distribution with
aeolian dust samples collected from the Ebinur drainage area (15-26 μm) (Ma et al.,
2016), further supporting the possible transport mechanism model proposed by us.
**5.2 Climatic events revealed by lake Ebinur sedimentary sequence**
The grain size record of lake Ebinur sediments reveals that the study area was in
a climatic transition stage from cold to warm during 1876-1920 AD (Figs. 8a-8h),
which is highly consistent with the temperature changes in China over the past 200
years reconstructed by Ge et al. (2013). In addition, the lacustrine sedimentary record
shows a marked change around 1955 AD (Figs. 8a-8h), which may be related to
regional human activity. On 1 October 1955, the Xinjiang Uygur Autonomous Region
was established, opening a new era of vigorous development in northwest China. The
sediments of lake Ebinur have become finer since 1955 (Figs. 8a, 8b), suggesting that
a decrease of coarse dust from local and/or regional sources, possibly due to the fixation
of surface dust by growing urbanization and intensive agricultural (Zhou, 1998). The
L* value increased continuously after 1955 (Fig. 8f), indicating the increase of
carbonate content, which may be caused by the rapid shrinkage of lake Ebinur. Since
the 1950s, intensive agricultural development in the lake Ebinur region, such as land
reclamation and irrigation, has led to a dramatic reduction in the lake's area and
increased aridity in the region (Ma et al., 2014; Zhang et al., 2015). Notably, Md and
EM2 show two abnormally high values since 1955 AD (Figs. 8a, 8b), and
correspondingly, the C values also show high values (Fig. 8c), indicating strong
transport dynamics (Jiang et al., 2017a; Shi et al., 2022). These two events (E1, E2)
may be related to local strong wind events within the age error (Wang et al., 2003).

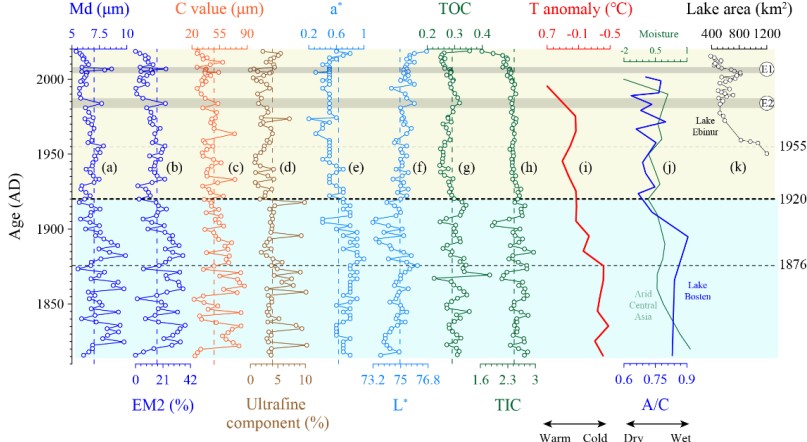




**Figure 8. Comparison of the multi-proxies record of sedimentary sequence in lake Ebinur with other climate records. (a) Median grain size (Md); (b) the percentage of EM2; (c) the one percent of grain size (C value); (d) the proportion of ultrafine component (< 1 μm fraction); (e) a\*; (f) L\*; (g) the total organic carbon content (TOC); (h) the total inorganic carbon content (TIC); (i) reconstructed China temperature anomaly (Ge et al., 2013); (j) blue line for pollen A/C ratios from lake Bosten (Chen et al., 2006a) and green line for synthesized moisture for arid central Asia (ACA) (Chen et al., 2010); (k) the area of lake Ebinur (Chen et al., 2006b; Maihemuti et al., 2020). The grey bars indicate two strong wind events.**

## 5.3 Climate transition and possible forcing mechanism

Clearly, multi-proxies analysis of lake Ebinur sedimentary sequence suggests that climate change over the 200 years can be divided into two periods by 1920 AD (Figs. 8a-8h). In the early period (1816-1920 AD), the climate of the study area was cold and wet, while it was warm and dry in the later period (1920-2019 AD). These results are consistent with the cold-wet and warm-dry climate combinations revealed by Chen et al. (2010, 2015) in arid central Asia. Moreover, the lake Ebinur sedimentary record reveals that a climate transition around 1920 AD, the same as the reconstructed temperature records in China (Yang et al., 2002; Ge et al., 2013), both of which indicate that China's LIA ended in 1920 AD.

Solar radiation (Lean et al., 1995; Wu et al., 2009)、volcanic eruptions (Gao et al., 2008; Brönnimann et al, 2019; Wang et al., 2022) and the concentrations of greenhouse gases ($CO_2$ and $CH_4$) (Mann et al., 1998; Jones and Mann, 2004; Huber and Knutti, 2011) are generally considered to be the main drivers of climate change. As shown in Figure 9, we collected data on total solar irradiance (TSI)、stratospheric sulfate injections from volcanic eruptions and the concentrations of greenhouse gases ($CO_2$ and $CH_4$) over the past 200 years for comparative analysis. The results show that during 1920-1950 AD, TSI increased significantly from 1360.4 W m$^{-2}$ to 1361.5 W m$^{-2}$ (Fig. 9b), while the concentrations of $CO_2$ (from 303.2 ppm to 312.6 ppm) and $CH_4$ (from 960.8 ppb to 1108.5 ppb) increased slowly (Fig. 9a). Since 1950 AD, the TSI has maintained a high value (Fig. 9b), and the concentrations of $CO_2$ (from 312.6 ppm to 409.5 ppm) and $CH_4$ (from 1108.5 ppb to 1681.6 ppb) have shown a rapid increase (Fig. 9a). Stratospheric sulfate injections from volcanic eruptions in the Northern Hemisphere have shown low levels since 1840s (Fig. 9b). Thus, we propose that the increase of TSI was the main controlling factor in the 1920 climate transition, and the gradual increase in the concentrations of greenhouse gases may have a positive feedback effect on the climate transition. In addition, the accelerated rise in the concentrations of greenhouse gases caused by human activity since 1950 AD (Fig. 9a), especially in the Xinjiang region (Zhou, 1998; Chen et al., 2006b), has further amplified the warming dominated by solar radiation.





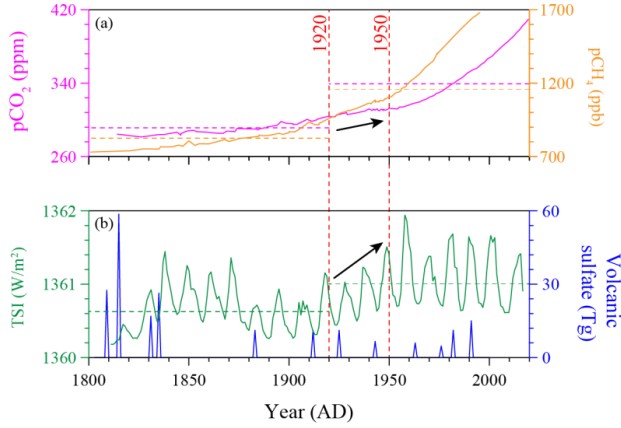


**Figure 9. Comparison between external climate forcing. (a) the concentrations of greenhouse gases: $CO_2$ (pink line) and $CH_4$ (orange line) (Keeling et al., 2005; Meure et al., 2006); (b) total solar irradiance (green line) (Lean, 2018) and stratospheric sulfate injections from volcanic eruptions (blue line) (Gao et al., 2008).**


## 6 Conclusions


The lake Ebinur sediments are mainly composed of fine-grained materials, and the median grain size ranges from 5.5 μm to 9.9 μm, with a mean value of 7.0 μm. Multi-parameter analysis of grain size suggests that the sediments are mainly transported by wind, and there are two kinds of different sources and transport processes: the fine-grained sediments (< 20 μm) are background dust that was transported by long distance high-altitude suspension, while the coarse-grained sediments (> 20 μm) are local and regional dusts that were transported from short distances at low altitudes. Based on the comparative analysis of grain size, color reflectance ($L^*$, $a^*$) and carbon content (TOC and TIC) of the lake Ebinur sedimentary sequence, we propose that the climate over the past 200 years can be divided into two periods by 1920 AD. In the early period (1816-1920 AD), the high C values indicate strong transport dynamics; and the high proportion of ultrafine component indicates strong pedogenesis, combined with high organic carbon content and high $a^*$ values, we inferred that the water vapor content is relatively higher. Thus, this period corresponds to the cold and wet climate. In the later period (1920-2019 AD), these proxies all show opposite changes, revealing a warm and dry climate. Through a comparative analysis of multiple climate-drivers, including total solar irradiance (TSI)、volcanic sulfate injections and the concentrations of greenhouse gases ($CO_2$ and $CH_4$), we conclude that the increase of TSI was the main controlling factor in the 1920 climate transition, and the gradual increase in the concentrations of greenhouse gases may have a positive feedback effect on the climate transition.


## Code/Data availability


All data can be obtained by contacting the author: xtwei@ies.ac.cn.


## Author contribution




XW undertook the laboratory analysis, created the figures, and drafted the paper. HJ, guided the writing, and modified the draft. HX and WS helped analyze data and optimize the draft. YL helped collected literatures. QG and SZ helped collect cores and fieldwork. All authors reviewed and approved the paper.

**Competing interests**

The authors declare that they have no conflict of interest.

**Acknowledgements**

This work was funded by the National Nonprofit Fundamental Research Grant of China, Institute of Geology, China Earthquake Administration (IGCEA1906).

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
