# Peer review of "Climate transition over the past two centuries revealed by"

_Climate of the Past, 2022_

## Referee Comment (RC2)

**Revision**

**Climate transition over the past two centuries revealed by lake Ebinur in Xinjiang, northwest China**

Xiaotong Wei, Hanchao Jiang, Hongyan Xu, Yumei Li, Wei Shi, Qiaoqiao Guo, Siqi Zhang

Dear Editor and Authors of the manuscript "Climate transition over 1 the past two centuries revealed by lake Ebinur in Xinjiang, northwest China",

Thank you for considering me to read this article and to be part of the process. Below, I am sending the revisions of the manuscript.

Sincerely,

**General comments**

In this manuscript, the authors provide a multiproxy study of a short sediment core from a shallow terminal lake in a climate-sensitive arid area. This record includes a significant number of variables, mainly based on particle size parameters, that can be linked to recent paleoenvironmental changes.

I read carefully the manuscript, and I was also able to read the discussion, the comments and their responses. Based on this information, I agree with the comments of the Editor and Referee 1 concerning the problems in the chronological model (I provide more details in the comments below). Unfortunately, I do not see that the authors provided enough evidence in their relplies to support the model. This is a major obstacle to be addressed when the goal of the paper is to provide a high temporal resolution of the sequence in order to determine the exact year of a climate shift. As another related issue, it is difficult to understand the criteria for the separation of units and subunits, and thus, the exact point of the sedimentation changes. It also surprises me that the changes on the lake area and water discharges during the deposition of this sequence are completely ignored in the discussion in the shallow lake sedimentation processes. Finally, I suggest to revise the organization of the manuscript separating the data from interpretation, and discussions, following a clear and logical order, and to provide better explanation of the figures.

Considering that some of the mentioned points intefere with the goals and the precision proposed in this work, my suggestion is to reject the manuscript. I hope that the authors can think about a solution to the main problems (maybe adding marker layers to the model),or to consider a different approach, to send the new manuscript using this valuable data.

**Specific comments**

**128-130. The climate of the study region is mainly dominated by westerlies and is a typical temperate continental climate, which is characterized by low rainfall and strong evaporation (Zhou et al., 2019, 2021).**

**138-139. The climate is generally characterized by warm-dry summers and cold wet winters (Fig. 2)**

Along the manuscript and in the reconstruction, the authors mention wet climate in the area…I do not think this term is adequate for this site, there are relatively wetter and drier stages but within the analyzed time range, the climate seems mostly arid.

**156-159...two parallel sediment cores (ABHHX and ABH, of 48 cm and 50 cm length, respectively) were retrieved from the northwest edge of lake Ebinur at a water depth of 0.8 m (45°04' N, 82°36' E, 193 m), using a 60-mm UWITEC gravity corer (Fig. 1)…**

Considering the location of the coring site (if the coordinates provided in the manuscript of the coring site are correct, **45°04' N, 82°36'**), the cores were taken close to the northwestern lake shore. The shallow water-depth and large area variations, immediately suggest that parts of lake bottom near the margins, such as the coring site, are susceptible to become air exposed. Moreover, historical satellite images from Google Earth show that the northern part of the lake was above the lake water level during images from the 80's and 90's. The same seems to be presented in maps of papers studying the lake area fluctuation (*Wang et al. 2021. Simulation of Lake Water Volume in Ungauged Terminal Lake Basin Based on Multi-Source Remote Sensing. Remote Sensing. 13.10.3390/rs13040697; Zhang et al. 2015. Environ Monit Assess 187:4128DOI 10.1007/s10661-014-4128-4)*. The lake bottom exposure and wind action of part of the 20[th] century sediments may also explain the lack of the [137]Cs signal in the Lake Ebinur sediments, which has been effectively detected in other lake sequences from the same region (*Lan, Jianghu, et al. "Time marker of 137Cs fallout maximum in lake sediments of Northwest China." Quaternary Science Reviews 241 (2020): 106413*), and in a different cores retrieved from the same lake at a greater water depth (*Liu et al 2017 J. Limnol., 2017; 76(3): 534-545*); *Ma L, Wu J, Abuduwaili J, Liu W (2016) Geochemical Responses to Anthropogenic and Natural Influences in Ebinur Lake Sediments of Arid Northwest China. PLoS ONE 11(5): e0155819. doi:10.1371/journal.pone.0155819*).  This should be carefully considered by the authors since it challenges the assumed "continuous and uniform sedimentation", and thus, questions the reliability of the age model (as pointed by RC1 and CE1), even when the mean particle size is similar along the sequence.

**171-176. The activity of total 210Pb (210Pbtot) was determined via gamma emissions at 44.5 keV, and the activity of 226Ra was determined by measuring the activity of its daughter nuclide 214Pb at 295 keV and 352 keV. The activity of 137Cs was measured with the 662 keV photopeak. The supported 210Pb activity was assumed to be in**

**equilibrium with in situ 226Ra activity, and the unsupported 210Pb activity (210Pbex) was calculated by subtracting the 226Ra activity from the 210Pbtot (Pratte et al., 2019).**

In the Materials and methods section, the authors indicate the methodology for calculating the 210Pb activity. Nevertheless, they do not indicate how they constructed the age model from this data, this should be addressed in the methodology. Additionally, it is indicated that the model applied is the CRS, which assumes a constant rate of supply and a variable sedimentation rate. For the oldest part of the sequence a linear extrapolation is assumed. I find an inconsistency in this point when the authors argue "uniform rates" (reply to Q5). Finally, in the replies to the comments, they say that "six data were discarded due to abnormal deposition rate". Where are these results? I think there is information missing from the manuscript.

**177-186.** In Materials and methods, the grain size distribution method is explained. However, the measured parameters are not mentioned. They should be mentioned in this section and explained in the Results and Interpretation section.

**188-191. Each Sample of about 1.5 g was dried at 40 °C for 24 h, then crushed without damaging their grain-size (Jiang et al., 2008) and the color reflectance was measured by using a SPAD 503 handheld spectrophotometer.**

The color reflectance methodology and the parameters' meanings are not explained.

**228-230. 4.2 Sedimentary Proxies record. All 96 grain-size data were analyzed by end-member analysis (EMA) using AnalySize software (Weltje, 1997; Paterson and Heslop, 2015; Jiang et al., 2022).**

This is methodological aspect, please move these lines to the previous section.

**231-239. Overall, the lower part of the sedimentary sequence is characterized by large fluctuations and coarse grains (Figs. 5a-5g), with over 60 % of the C value above 50 μm (Figs. 6b, 6c), reflecting strong transport dynamics (Passega, 1964; Jiang et al., 2017a; Wei et al., 2021). The particles in the upper part of the sequence are finer (Figs. 5a-5g), and only ~ 20 % the C value exceeds 50 μm (Figs. 6d, 6e), indicating smaller transport dynamics.**

 **In addition, other proxies of lake Ebinur sequence (L\*, a\*, TOC and TIC) also show different variation characteristics in the corresponding upper and lower parts (Figs. 5k-5n). Thus, the sedimentary sequence can be divided into two units.**

In this description, the "upper part" and the "lower part" of the sedimentary sequence are not well delimited. Furthermore, the criteria for the separation of the units and subunits should be

clearly defined. This is important since the contact between unit 1 and unit 2 is established as the point where sedimentation and climate shifted.

**Figure 5.** This Figure includes some parameters that are not explained in the manuscript, or sometimes they are not explained, interpreted nor used along the manuscript (e.g. the Mode, EM1, <2 micrometers, etc.).

**258-260. The variation of Md is clearly influenced by the coarse component: EM2 (0-37.6 %, mean 21.6 %) (Fig. 5g), i.e., the coarse particles are deposited first, and the fine particles are deposited later.**

From the Figure 5, I cannot detect differences in excursions of the curves from EM1 and EM2, they seem quite similar. Please, explain them in the results.

**260-263. Combined with the higher C value for this unit (22.6-86.5 µm, mean 54.5 µm) (Fig. 5i), it indicates that the wind is stronger at this stage, bringing more coarse-grained matter from local and regional dust (see Sect. 5.1 for the explanation of provenance).**

Have the authors considered the role water inflow in sediment transport in any part of the sequence?

**266-269. Correspondingly, the contents of TOC (0.25-0.34, mean 0.30) and TIC (1.91-2.95, mean 2.50) also showed strong fluctuations (Figs. 5m, 5n), which may have been influenced by strong wind activity during the cold period.**

Please, provide an explanation of this relationship.

**272-274. In this unit, the proportion of ultrafine component is the highest in the whole sequence (1.7 %-10.2 %, mean 5.0 %), revealing the strongest pedogenesis in the study area (Fig. 5h).**

How do authors determine that the ultrafine component is local? Could it be part of the long-distance transport?

**285-287. Related to humidity fluctuation, L\* values within arid lakes are considered to reflect variations in the carbonate, and high L\* values denote more carbonate content (Xiao et al., 2006; Jiang et al., 2008).**

According to Figure 5, L* and TIC curves seem to fluctuate quite differently. Moreover, L* seems lower in unit 2 than in unit 1. Therefore, I think that the interpretation of the *L and TIC values needs further explanation.

**294-298. Further, this unit can also be divided into two sub-units based on the changes of all proxies: unit 1-a and unit 1-b (Figs. 5a-5n). During the period of unit 1-a (48-34 cm, 1816-1876 AD), the study area has been under cold and wet climatic conditions, while unit 1-b (34-24 cm, 1876-1920 AD) was in a transition period from cold and warm.**

This is not sustained by any explanation in this section, it seems an arbitrary division. The same is applicable to the following subunits. Please, provide more details.

**Figure 6**

This Figure is not mentioned or explained in the manuscript.

**310-313. This is probably duo to the decrease of the temperature gradient as the temperature rise (Zhang et al., 2021), resulting in the weakening of wind intensity and the decrease of coarse particles transported at low altitude and short distance (Ge et al., 2016).**

I saw that there was an explanation added in the response to a question about the meaning of the temperature gradient. I find that the relation of temperature changes and the grain size in this lake record is vague and should be better addressed in the discussion.

**313-314. As well, the contents of TOC (0.25-0.32) and TIC (2.17-2.63) showed very slight fluctuations except for the top two points (Figs. 5m, 5n).**

What does this mean?

**329-334. 5.1 Provenance and transport mechanisms of lake Ebinur sediments. The Y value of Sahu's formula is usually used to recognize the eolian environment, which is mainly determined by mean grain size, standard deviation, skewness, and kurtosis (Sahu, 1964). The Y values of all samples range from -19.5 to -7.6, lower than the threshold value of -2.74 (Fig. 7), supporting their windblown origin (Jiang et al., 2017b, 2022; Wei et al., 2021).**

The results of the end members and this interpretation should be in the Results and Interpretation section, before authors interpret the whole sequence. In my opinion, provenance is not analyzed in this section. In addition, please, provide further explanation

about the formula and threshold value (is this number and formula used to recognize only the aeolian origin or to discriminate aeolian from beach transport, as in Sahu, 1964?).

354.     **5.2 Climatic events revealed by lake Ebinur sedimentary sequence**

In this section, authors include also the human influence on the lake sediments, so it would be convenient to modify the title.

**364-366. The L\* value increased continuously after 1955 (Fig. 8f), indicating the increase of carbonate content, which may be caused by the rapid shrinkage of lake Ebinur.**

This is contradictory with the rest of the interpretations of carbonate precipitation, since unlike in the underlying units, here, carbonate precipitation is interpreted as triggered by lake volume reduction. Please, explain the mechanisms of carbonate precipitation in this lake.

**369-373. Notably, Md and EM2 show two abnormally high values since 1955 AD (Figs. 8a, 8b), and correspondingly, the C values also show high values (Fig. 8c), indicating strong transport dynamics (Jiang et al., 2017a; Shi et al., 2022). These two events (E1, E2) may be related to local strong wind events within the age error (Wang et al., 2003).**

Please, indicate the ages and the errors of these two events. Are these events regional? Can they be tracked in different cores or in other records from the region? Could they be used as time markers?

**426-429. The fine-grained sediments (< 20 µm) are background dust that was transported by long distance high-altitude suspension, while the coarse-grained sediments (> 20 µm) are local and regional dusts that were transported from short distances at low altitudes.**

This is not analyzed in the results, in the interpretations or in the discussion.

385.     **5.3 Climate transition and possible forcing mechanism. Figure 9**

In this comparative Figure, the main data obtained from of this work should be included.

---

## Author Comment (AC1)

Figure S1. Age-Depth models of the cores ABH and ABHHX based on the CRS model

[Figure]

Figure S2. Linear extrapolation age of the core ABHHX

[Figure]

Formula (1)

$$t = \frac{1}{\lambda} ln \frac{A_{(0)}}{A}$$ (Appleby and Oldfield, 1978)

$\lambda$: the decay constant of $^{210}$Pb (0.0311 yr$^{-1}$); $A_{(0)}$: the total, integrated $^{210}$Pbex activity in the core ($\sum^{210} Pbex$); A: the cumulative $^{210}$Pbex below the level representing time t.

Formula (2)

TOC $_{act.}$ = TOC $_{mea.}$ * (sample weight after carbonate removal / sample weight before carbonate removal)